# Genomic Analysis of Shiga Toxin-Producing *E. coli* O157 Cattle and Clinical Isolates from Alberta, Canada

**DOI:** 10.3390/toxins14090603

**Published:** 2022-08-31

**Authors:** Emmanuel W. Bumunang, Rahat Zaheer, Kim Stanford, Chad Laing, Dongyan Niu, Le Luo Guan, Linda Chui, Gillian A. M. Tarr, Tim A. McAllister

**Affiliations:** 1Faculty of Veterinary Medicine, University of Calgary, Calgary, AB T2N 1N4, Canada; 2Agriculture and Agri-Food Canada, Lethbridge Research and Development Centre, Lethbridge, AB T1J 4B1, Canada; 3Department of Biological Sciences, University of Lethbridge, Lethbridge, AB T1K 1M4, Canada; 4National Centre for Animal Disease Canadian Food Inspection Agency, Lethbridge, AB T1J 0P3, Canada; 5Department of Agricultural, Food and Nutritional Science, University of Alberta, Edmonton, AB T6G 2P9, Canada; 6Alberta Precisions Laboratory, Alberta Public Health, Edmonton, AB T6G 2J2, Canada; 7Department of Laboratory Medicine and Pathology, University of Alberta, Edmonton, AB T6G 2B7, Canada; 8Division of Environmental Health Sciences, School of Public Health, University of Minnesota, Minneapolis, MN 55455, USA

**Keywords:** *Escherichia coli* O157, Shiga toxins, *stx*-carrying phages, insertion sites, antimicrobial resistance

## Abstract

Shiga toxin (*stx*) is the principal virulence factor of the foodborne pathogen, Shiga toxin-producing *Escherichia coli* (STEC) O157:H7 and is associated with various lambdoid bacterio (phages). A comparative genomic analysis was performed on STEC O157 isolates from cattle (*n* = 125) and clinical (*n* = 127) samples to characterize virulence genes, *stx*-phage insertion sites and antimicrobial resistance genes that may segregate strains circulating in the same geographic region. In silico analyses revealed that O157 isolates harboured the toxin subtypes *stx1a* and *stx2a.* Most cattle (76.0%) and clinical (76.4%) isolates carried the virulence gene combination of *stx1*, *stx2*, *eae* and *hlyA*. Characterization of *stx1* and *stx2*-carrying phages in assembled contigs revealed that they were associated with *mlrA* and *wrbA* insertion sites, respectively. In cattle isolates, *mlrA* and *wrbA* insertion sites were occupied more often (77% and 79% isolates respectively) than in clinical isolates (38% and 1.6% isolates, respectively). Profiling of antimicrobial resistance genes (ARGs) in the assembled contigs revealed that 8.8% of cattle (11/125) and 8.7% of clinical (11/127) isolates harboured ARGs. Eight antimicrobial resistance genes cassettes (ARCs) were identified in 14 isolates (cattle, *n* = 8 and clinical, *n* = 6) with streptomycin (*aadA1*, *aadA2*, *ant(3’’)-Ia* and *aph(3’’)*-*Ib*) being the most prevalent gene in ARCs. The profound disparity between the cattle and clinical strains in occupancy of the *wrbA* locus suggests that this trait may serve to differentiate cattle from human clinical STEC O157:H7. These findings are important for *stx* screening and *stx*-phage insertion site genotyping as well as monitoring ARGs in isolates from cattle and clinical samples.

## 1. Introduction

Shiga toxin-producing *Escherichia coli* (STEC), especially O157:H7, is an important food and waterborne pathogen. Cattle are considered asymptomatic carriers of STEC O157:H7 and food products and surface/ground water contaminated with cattle feces can serve as vehicles of transmission to humans and other animals. STEC O157:H7 outbreaks associated with beef [1] and pork products [2] have been reported in Alberta. This corroborates the evidence that people living in areas with large numbers of cattle are at higher risk of STEC infections [3,4]. One possible explanation for this phenomenon is that particular strains persist in cattle and their environment, occasionally crossing over to humans and causing small, local outbreaks and sporadic infections.

Shiga toxin (Stx) is STEC’s principal virulence factor, encoded by *stx1* and *stx2* genes and is often associated with lambda bacteriophages [5] which insert into the bacterial chromosome at specific sites. Some of these insertion sites include *wrbA*, *yecE*, *sbcB*, *argW*, *Z2755*, *torS/T*, *ynfH*, *yciD*, *potC*, *serU*, *yjbM* and *mlrA* [6,7]. Occupation of these insertion sites by different prophages contributes to diversity within STEC [8,9] as well as virulence [10]. For instance, *stx1-* and *stx2*-carrying prophages are inserted at *mlrA* and *wrbA* insertion sites, respectively, in *E*. *coli* O157:H7 Sakai [5], a strain associated with an outbreak from contaminated radish sprouts in Japan [11]. The *stx*-carrying φPOC-J13 phages can infect clinical strains such as *Shigella flexneri* [12,13] and *S*. *sonnei* [14]. These *stx*-carrying φPOC-J13 phages contribute to the genetic heterogeneity of the *stx*-carrying phage pool in humans and cattle. Thus, a differential diagnostic method which combines stool culture and *stx* detection is necessary to distinguish between STEC- and non-STEC-related infections [15] for outbreak surveillance purposes.

Stx can bind to globotriosylceramide (Gb3) receptor cells [16] which are located in the human intestinal tract, kidney and brain, causing damage to these organs usually characterized by bloody diarrhoea, thrombocytopenia, haemorrhagic colitis and hemolytic uremic syndrome (HUS) [17]. Symptoms may vary depending on the virulent nature of the strain, immunological condition of the patient and age [15]. Stx are usually transported in blood to these target organs through polymorphonuclear leukocytes and have a half-life of about 5 days in blood [18], an important etiologic factor for diagnosis and treatment as these toxins are undetectable in stools after 5 days [18]. Supportive therapy with hydration to maintain electrolyte balance is the most reliable method of treating STEC infections in humans. Antibiotic treatment of STEC O157 is not usually recommended in humans due to concerns that it may increase toxin production [19,20,21]. Some DNA-based vaccines that are capable of neutralizing *stx* activity [22] and STEC shedding [23] in mice are a promising strategy but have yet to undergo human trials. Polyphenolic compounds such as condensed tannins and phlorotannins, which can inhibit STEC growth [24] could be promising therapeutic agents in humans that can help reduce STEC growth and subsequent attachment in the intestinal tract. Furthermore, *Weissella confuse*, a probiotic which has shown activity against STEC in a zebrafish model [25] and monoclonal antibodies that can clear *stx2* in the bloodstream of mice [26] all have potential as therapeutic agents in humans, if clinical efficacy can be confirmed.

STEC strains are mosaic in their virulence profiles locally and globally, however, certain lineages may persist in the environment and circulate between animals and/or humans. For example, STEC O157 sequence type eleven (ST11) is a predominant clone [27,28] circulating in humans and cattle that has been isolated from human, bovine and porcine samples in Alberta [29,30,31]. STEC O157 infections in Alberta have been associated with HUS, primarily in children <10-years-old and the elderly >60-years-old [32].

In addition to *stx* genes, other virulence factors such as *eae* (intimin), which are essential to the locus of enterocyte effacement (LEE), translocated intimin receptor gene (*tir*), *esc* cluster genes that encode for components of the type III secretion system responsible for secretion of Esp proteins (coded by *espA*, *espB* and *espD*) and hemolysin (*hylA*) are associated with intestinal and kidney disease in humans [33].

Antimicrobial agents that are used to control or treat infections in humans, livestock, and crops can promote carriage of antimicrobial resistance (AMR)/resistance genes (ARGs) in bacteria. Antimicrobial resistant-bacteria are considered adulterants in food producing animals and pose a threat to global public health, food security and economic growth [34]. STEC O157 strains resistant to streptomycin, sulfisoxazole, and tetracyclines are commonly associated with isolates from commercial feedlots [35] and clinical [36,37] samples in Canada and abroad [38,39]. Mobile DNA elements such as transposons and plasmids are vehicles and major distributors of ARGs through horizontal gene transfer within or across bacterial species.

Whole genome sequencing (WGS) is increasingly being used by the Centers for Disease Control and Prevention, the Food and Drug Administration, the United States Department of Agriculture’s Food Safety and Inspection Service [40] and the Public Health Agency of Canada [41] for surveillance and to discriminate closely related STEC from outbreak events. STEC are prevalent and highly diverse in cattle [42], while strains that cause severe human disease are less diverse and infrequent [43]. Furthermore, there is growing evidence of secondary transmission of STEC O157:H7 [44,45], which suggest that humans can act as carriers. Therefore, to understand STEC O157 strains circulating within the same geographic region in cattle and humans in Alberta, we sequenced STEC O157 for a comparative genomic analysis of clinical isolates collected in hospitals and isolates collected from feedlot cattle from 2007 to 2015.

## 2. Results

### 2.1. In Silico Serotyping and Multi-Locus Sequence Typing (MLST) Analysis

Two hundred and fifty-two bacterial isolates from cattle (*n* = 125) and humans (*n* = 127) were sequenced. In silico serotyping revealed that all clinical STEC O157 isolates possessed the H7 antigen determinant, whereas 79.2%, 4.0%, 4.8% and 4.0% of cattle isolates had H7 (*n* = 99), H12 (*n* = 5), H19 (*n* = 6) and H29 (*n* = 5), respectively (Appendix A). These O157:non-H7 (*n* = 16) were detected only in cattle samples. MLST analyses revealed five different sequence types (STs). Most cattle (87.2%) and clinical isolates (98.4%) belonged to ST11. Cattle isolates also included ST10 (*n* = 5), ST515 (*n* = 4), ST763 (*n* = 6) and ST9964 (*n* = 1). The ST of two clinical isolates could not be identified. Overall, 98.4% (125/127) of clinical and 87.2 (109/125) of cattle isolates were ST11 (Figure 1A).

### 2.2. Distribution of Stx Genes, Prophages, and Stx-Phage Insertion Sites

The majority of cattle and clinical isolates carried *stx1/2* loci in tandem that encoded A and B toxin subunits that were identical to the *stx1/2* in *E**. coli* O157:H7 strain Sakai (NC002695.2) [5] and EDL933 (CP008957.1) [46]. In *E**. coli* O157 genomes of both cattle and clinical origin, only *stx1a* and *stx2a* gene subtypes (referred in this study as *stx1* and *stx2*) were identified. These genes showed 100% nucleotide sequence similarity across cattle and the clinical isolates. The *stx1* and *stx2* genes were common in both cattle (76.0%; *n* = 95, 84.8%; *n* = 106) and clinical (76.4%; *n* = 97, 100%; *n* = 127) isolates, with the majority of cattle (76.0%) and clinical (76.4 %) isolates possessing both *stx1* and *stx2* (Figure 1b). The O157:non-H7 (*n* = 16) lacked *stx1*/*stx2*.

Identified prophages were distributed between cattle (intact; *n* = 278, questionable; *n* = 223 and incomplete *n* = 951) and clinical (intact; *n* = 287, questionable; *n* = 228 and incomplete *n* = 1096) isolates, with incomplete prophages being the most prevalent (Figure 1C). Manual screening of assembled contigs for *stx1* and *stx2*-carrying phages revealed that the integrase (*int*) gene was present at *mlrA* or *wrbA* insertion sites (Figure 2A) and indicative of phage occupation whereas these regions were unoccupied (absence of phage *int*) in some isolates (Figure 2B). The O157:non-H7 (*n* = 16) cattle isolates that lacked both *stx1*/*stx2* were not considered for insertion site comparative analysis. Insertion sites were more occupied in cattle (*mlrA*, *n* = 97 and *wrbA*, *n* = 99) than clinical (*mlrA*, *n* = 49 and *wrbA*, *n* = 2) isolates (Figure 3). Of the 49 clinical isolates having occupied *mlrA* loci, 30.6% (15/49) lacked *stx1* in their genome compared to 3.1% (3/97) in cattle (Figure 3). Overall, all the unoccupied *mlrA* regions in cattle (*n* = 12) and clinical (*n* = 15) isolates lacked *stx1* (Figure 3) and *int* in their genomes. Contrarily, 100% (19/19) of clinical isolates with unoccupied *wrbA* loci, possessed *stx2*, whereas 83.3% (5/6) of cattle isolates had *stx2* (Figure 3). With the exception of *argW*, no other *stx2*-insertion site such as *sbcB* or *yecE* were occupied by prophages (data not shown). Those prophages identified within the *argW* insertion site were not stx-phages.

Sequence polymorphism (33 bp nucleotide deletions and 18 bp nucleotide insertions) were identified when *wrbA* within unoccupied clinical (83.5%, 106/127 and 14.9%, 19/127) and cattle isolate (3.2%, 4/125 and 4.8%, 6/125) sites were aligned against that of *E**. coli* O157:H7 strain Sakai (Figure 4). The 18 bp insertions in *wrbA* did not correspond to phage sequence, but rather to an intergenic sequence that flanked *wrbA* within isolates with occupied stx2 prophages.

### 2.3. Additional Identified Virulence Factors

Additional virulence genes identified included Type II secretion protein (EtpD), essential genes of the locus of enterocyte effacement (LEE) intimin (*eae*), translocated intimin receptor gene (*tir*), esc cluster genes that encode for the type III secretion system responsible for secretion of Esp proteins (coded by *espA*, *espB* and *espF*), prophage-encoded type III secretion system effector (*espJ*), extracellular serine protease plasmid-encoded (*espP*), hemolysin (*hylA*), EAST-1 heat-stable toxin (*astA*), colicin encoded virulence factor (*cib*, *cia* and *cba*), glutamate decarboxylase (*gad*), plasmid-encoded catalase peroxidase (*katP*), Toxin B (*toxB*), increased serum survival (*iss*), outer membrane hemin receptor (*chuA*), Non-LEE encoded effector (NleA, NleB and NleC), outer membrane protease (OmpT), outer membrane complement resistance protein (TraT), tellurium ion resistance protein (TerC) and adherence protein (Iha) (Appendix A). In both cattle and clinical isolates carrying either *stxt1*+ /*stx2*+ or *stx1*−/*stx2*+, additional virulence genesincluding *eae*, *hylA*, *tir*, *espA*, *espB*, *espP*, *espJ*, *katP*, *toxB*, *nleABC* and *chuA* were always present, followed by *iha* (clinical *n* = 127 and cattle *n* = 110), *espS* (clinical *n* = 110 and cattle *n* = 39), *cia* (clinical *n* = 4 and cattle *n* = 9), *cba* (clinical *n* = 2 and cattle *n* = 5) and *cib* (cattle *n* = 3). Sixteen cattle isolates with the *stx1*−/*stx2*− profile lacked *eae*, *hylA*, *tir*, *espA*, *espB*, *espP*, *katP*, *toxB*, *nleABC* and *chuA*, with *terC* being the only virulence gene present in all O157 isolates.

### 2.4. Predicted Antimicrobial Resistance/Resistance Cassette Genes and Plasmids

ResFinder revealed that 8.8% of cattle (11/125) and 8.7% of clinical (11/127) isolates harboured ARGs. Among ARG-carrying isolates, 19 isolates (clinical, *n* = 10 and cattle, *n* = 9) harboured between 2 to 10 ARGs (Table 1). Six of the 11 cattle isolates were non-STEC (lacked both *stx1*/*stx2*) whereas all 11 clinical isolates had either *stx2* or both *stx1*/*stx2*. The predicted phenotypes included resistance against ampicillin, streptomycin, kanamycin/neomycin, chloramphenicol, trimethoprim, sulfisoxazole and tetracycline. The most prevalent multidrug resistance (MDR) gene combinations included *ant(3’’)-Ia*, *sul1*, *tet(A)*, (*n* = 5 isolates) associated with 4 cattle and 1 clinical isolate, followed by *aph(6)-Id*, *aph(3’’)-Ib*, *sul2*, *tet(B)*, which were associated with 3 clinical isolates. Collectively, aminoglycoside resistance genes were most prevalent (*n* = 28), followed by tetracycline (*n* = 26), sulfonamide (*n* = 17), phenicol (*n* = 5), beta-lactam (4), and trimethoprim (*n* = 2). Beta-lactam resistance genes were only present in clinical isolates. Eleven clinical isolates with ARGs carried virulence gene combination (*stx1^+^*/*stx2^+^* and *stx1^−^*/*stx2^+^*) compared to five cattle isolates (*stx1*+/*stx2*+ and *stx1*−/*stx2*+) (Table 1).

Eight antimicrobial resistance genes cassettes (ARCs) were associated with transposase and integron-integrase in 14 isolates (cattle, *n* = 9 and clinical, *n* = 5) and streptomycin resistance (*aadA1*, *aadA2*, *ant(3’’)-Ia* and *aph(3’’)-Ib*) was most prevalent in different ARCs followed by the sulfonamide resistance (*sul2*) (Figure 5). Two clinical isolates with beta-lactam resistance genes (*blaTEM-1C* and *blaTEM-1B*) were flanked by transposase as well as one cattle isolate (Figure 5; ARC1, 2 and 3). Meanwhile, three clinical isolates (Figure 5; ARC4, *n* = 1 and ARC5, *n* = 2) and nine cattle isolates (Figure 5; ARC4, *n* = 4, ARC6, n = 1, ARC7, *n* = 3 and ARC8, *n* = 1) were flanked by an integron-integrase gene (class 1). ARC4 was the only ARC profile detected in both clinical (*n* = 1) and cattle (*n* = 4) isolates. ARC9, a non-integrated ARC was detected in three clinical isolates.

Twenty-seven different types of small plasmid replicons <1 kb (63–885 bp) were detected in both cattle and clinical isolates (Appendix A). The most prevalent was IncFII (*n* = 245), followed by IncFIB(AP001918) (*n* = 242), IncFIA (*n* = 16), IncI1 (*n* = 16), pEC4115 (*n* = 13) and IncI2 (*n* = 8). Plasmid replicons, Col(BS512) (*n* = 1), ColRNAI (*n* = 2), IncA/C2 (*n* = 1), IncFIA(HI1) (*n* = 1), IncFIB(K) (*n* = 1), IncFIC(FII) (*n* = 6), IncHI2 (*n* = 1), IncHI2A (*n* = 1), IncX4 (*n* = 4) and IncY (*n* = 4) were unique to cattle isolates, whereas Col440I (*n* = 2), ColpVC (n =1), IncB/O/K/Z (*n* = 2), IncFII(pCoo) (*n* = 1) IncN (*n* = 1), p0111 (*n* = 1) and pXuzhou21 (*n* = 1) were associated with clinical isolates. Overall, the Inc (*n* = 19) types were the most prevalent followed by colicinogenic (*n* = 5) plasmids (Appendix A).

## 3. Discussion

This study conducted a comparative analyses of STEC O157 from cattle and clinical samples in the same geographic region. Our findings revealed evidence of similar and dissimilar virulence profiles in STEC O157 strains based on MLST, *stx* and *stx*-phage insertion site genotyping. In addition, mobile DNA elements such as transposons and integrons, which are major contributors to genetic variation and drivers of antimicrobial resistance among bacteria, were identified. In silico serotyping revealed that clinical (*n* = 127) and cattle (*n* = 99) isolates with O- and H-antigen determinant genes also have a *stx1*+, *stx2*+, *eae*^+^ and *hly*^+^ profile confirming their pathogenicity potential as O157:H7. However, sixteen O157:non-H7 cattle isolates H19; (*n* = 6), H29; (*n* = 5) and H12; (*n* =5) were considered non-STEC and may not be pathogenic as they all lacked *stx*1^−^, *stx*2^−^, *eae*^−^ and *hly*^−^. Based on whole genome (wg) MLST, the most prevalent O157:H7 STEC clonal lineage circulating in cattle and humans belong to the same sequence type, ST11 as the O157:H7 pathogen type [27] and corroborates previous *E. coli* O157:H7 studies in Alberta which found that ST11 was the predominant clone among cattle isolates [30,31]. Most ST11 isolates in both cattle and clinical samples possessed a similar virulence gene profile (*stx*1^+^, *stx*2^+^, *eae*^+^ and *hly*^+^), suggesting that wgMLST is a good predictor of isolates from cattle that may cause clinical disease in humans. Similarly, ST10, a non pathogen based on O157:H7 sequence typing [27] and other STs 515; (*n* = 4), 763; (*n* = 6) and 9964; (*n* = 1) in cattle isolates, had similar non-pathogenic (*stx1*−, *stx2*−, *eae*^−^ and *hly*^−^) profiles. Although wgMLST can distinguish potential pathogenic and non-pathogenic strains and may have a high discriminatory power compared to phenotypic typing methods, the discriminatory power of this method should be interpreted with caution or used in conjunction with other virulence profiling methods as it failed to predict the ST for two clinical isolates (*stx1*−/*stx2*+) in this study. These could be new, uncharacterized STs that possess a different genetic repertoire compared to those STs that are reported in the *E. coli* MLST [27] database.

The *stx*1/2 including subtypes are classified based on difference in protein sequence and biological activity [47]. Toxicity and cell receptor binding affinity of the different *stx* types play a major role in clinical outcomes [48] with *stx*2a and *stx*2c more potent than *stx*1, *stx*1c, sx2d and *stx*2e in humans [49,50,51,52]. The *stx*1a/2a subtypes identified in this study have previously been reported in cattle [31] and clinical [29] isolates from Alberta and suggest these are the main subtypes circulating in this region. This indicates a possible risk of *E. coli* O157:H7 circulating between cattle and humans given that human to human transmission should be less common in high-resource countries due to more robust sanitation practices. A combination of *stx*2a and *eae* genes in O157:H7 maybe associated with hemolytic uremic syndrome [29]. The majority of cattle and clinical isolates in this study possessed both *stx*2a and *eae* genes which could be indicative of their virulence potential in humans.

The *stx*1 and *stx*2 genes are carried by two different prophages which can infect the same bacterial strain. In this study, fragmented assemblies failed to reveal intact *stx*1/2-carrying prophages in O157:H7 genomes. Therefore, two Sakai phages (Sp) Sp5 and Sp15 from *E. coli* O157:H7 Sakai which carry *stx*2 and *stx*1, respectively, were used as reference *stx*-carrying phages to search for corresponding gene sequences in O157:H7 isolates. The int gene, which catalyses phage integration in Sakai phage had 100% sequence identity to that in O157:H7 from Alberta, illustrating the conservation of this phage within this serotype.

*Stx*-carrying phages in O157 *E. coli* are important in the dissemination of *stx* genes and genetic diversity of the bacterial host. Although most cattle and clinical isolates showed a similar genotypic profile by wgMLST and *stx* typing, this was not the case for *stx*1/2-carrying-phage insertion genotyping. There was a substantial disparity in occupied/unoccupied *wrbA* insertion site between cattle and clinical isolates, which raises the question as to why this site was occupied by phage in cattle, but not in clinical isolates. According to Serra-Moreno et al. [53], insertion site occupancy by *stx*-carrying phage is host strain and locus specific. Fourteen cattle isolates with *stx1*−/*stx2*− profile expectedly lacked both *mlrA* and *wrbA* insertion sites, whereas most clinical isolates (*n* = 95) unexpectedly had unoccupied *wrbA*. Differential occupation of *stx*2-phages in clinical isolates compared to cattle isolates suggests that O157:H7 strains circulating in humans and cattle differ, even though cattle are seen as the main reservoir of this pathogen. This may serve as an important epidemiologic trait which could differentiate these strains during outbreak events. This finding is supported by that of Shaikh and Tarr [9] who found that O157:H7 isolates from several sources including hamburgers had unoccupied *wrbA* locus. However, most isolates in the study of Shaikh and Tarr [9] were *stx1*−/*stx2*+, contrary to our findings as most O157:H7 clinical isolates (*n* = 95) were *stx1*+/*stx2*+.

To further understand the disparity in the unoccupied *wrbA* locus in clinical isolates, we looked for sequence polymorphisms between the unoccupied O157:H7 isolates’ *wrbA* locus and that of *E. coli* O157:H7 Sakai. A 33 bp sequence deletion or an 18 bp insertion in the *wrbA* gene were identified, modifications that may be responsible for the lack of occupancy in most clinical as compared to cattle isolates. An altered and unoccupied locus, indicates that selection of secondary insertion site is not only limited to absence of primary site [53], but also possibly to modification of the insertion sequence. Therefore, we hypothesized that the presence of *stx*2-carrying phages in O157:H7 with an unoccupied and a defective *wrbA* locus may be an adaptive response by phages to overcome a possible host-associated defense mechanism through integration at a different site in the O157:H7 chromosome. Fragmented assemblies failed to reveal flanking genes (insertion site) of the phage integrase of these isolates with unoccupied and or a defective (33 bp deletion/18 bp insertion) *wrbA* site. Further, the phage *int* gene in *stx*2-carrying phages in isolates with defective *wrbA* locus was 100% identical to that of Sp15, ruling out the possibility that changes in the sequence of *int* were responsible for the lack of integration at this site.

Other *stx*2-carrying phage integration sites in O157:H7 such as *argW*, *sbcB*, and *yecE* have been reported [6,54] and phage may prefer these insertion sites when the primary *wrbA* site is absent [53]. Except for *argW*, these integration sites were unoccupied in this study. However, the *int* at the *argW* site was identical to integrase of Sp16, a non *stx*-carrying phage in *E. coli* O157:H7 strain Sakai. The fact that a 33 bp sequence deletion is limited to most (*n* = 95) *stx*2-carrying clinical isolates with the *stx*1^+^/*stx*2^+^ profile, suggests they are a common clone that circulates in humans and may be genotypically different from those in cattle (*n* = 90) with the same *stx* (*stx*1^+^/*stx*2^+^) profile. Only 4 cattle isolates with unoccupied defective *wrbA* had the same *stx* (*stx*1^+^/*stx*2^+^) profile as those of clinical isolates, possibly a reflection of a similar lineage that circulates in both cattle and humans.

Few (clinical, *n* = 15/19 and cattle, *n* = 3/6) isolates with 18 bp sequence insertion in the unoccupied *wrbA* locus harboured *stx*1^−^/*stx*2^+^ in their genome. In the study of Shaikh and Tarr [9], these *stx*1^−^/*stx*2^+^ isolates lacked the duplicated GACATATTGAAAC intergenic sequences that flanked the *wrbA* insertion site. However, we found that, the terminal repeat sequence GACATATTGAAAC and a 5 bp (GTACG) sequence was unexpectedly fused (18 bp insert) within the *wrbA* gene of unoccupied O157:H7 strains (Figure 4). Thus, the O157:H7 STEC in this study with altered *wrbA* locus (18 bp insert) are genotypically different to unoccupied O157:H7 STEC with a 33 bp deletion as well as an occupied intact *wrbA* site. Shaikh and Tarr [9], suggest that O157:H7 STEC with *stx*1^−^/*stx*2^+^ profile might be the remnants of O157:H7 that gave rise to *stx*1^+^/*stx*2^+^ O157:H7 or were previously *stx*1^+^/*stx*2^+^ O157:H7 that lost their *stx*1 gene.

Interestingly, we also observed that clinical, 30.6% (15/49) and cattle, 4.1% (4/98) isolates with an occupied *mlrA* loci, lacked *stx*1 in their genome. A possible explanation for this is the partial loss of the *stx*1-carrying phage segment. For example, the remnants of integrated phage *int* gene were still present in the occupied site in agreement with Shaikh and Tarr [9] who reported the absence of *stx*1 in most O157:H7 strains with an occupied *mlrA*. Shiga toxin loss is an observation reported in O157:H7 [30,55] and is strain- and *stx* type-related [56]. However, the absence of *stx*1 was more prevalent in clinical isolates than cattle and suggest it could be host-associated. It has been shown that the remnants of *stx*1-carrying phage segments in O157 have the propensity to recombine with other mobile genetic elements or acquire and disseminate the *stx*1 gene [57]. Consequently, defective *stx*1-carrying phages in O157:H7 strains in this study may reflect a cycle of loss and gain of these elements.

Other than *stx* genes, most cattle and clinical isolates with *stx*1^+^/*stx*2^+^ or *stx*1^−^/*stx*2^+^ profile possessed a similar virulence gene profile which included Type III secretion LEE encoded Esp effector proteins, tir receptor protein, intimin protein, toxin B and associated adherence protein factors common to STEC O157 [58,59]. These factors, especially *eae*, detected in all *stx*1^+^/*stx*2^+^ or *stx*1^−^/*stx*2^+^ are important for STEC adherence to intestinal epithelia cells during infection in humans.

The number of isolates with antimicrobial resistance genes (ARGs) were equally distributed (11 each) in cattle and clinical samples with different resistance/multidrug profiles, possible reflective of differences in antimicrobial use in humans vs. cattle. Although ARGs were detected in low numbers in cattle (8.8%; 11/125) and clinical (8.7%; 11/127) isolates, most genes were associated with antimicrobials of clinical importance, including category II antimicrobials such as aminoglycosides and penicillins [60].This category was also more prevalent in clinical isolates with *stx*1^+^/*stx*2^+^ or *stx*1^−^/*stx*2^+^ profiles than cattle. Additionally, ARGs were flanked by transposase and integron-intergrase genes, which are mobile elements that can facilitate the dissemination and prevalence of ARGs horizontally and vertically within O157:H7 or other strains. The integron-integrase detected in this study was a class 1 integron common to the *Enterobacteriaceae* [61,62]. The streptomycin resistance gene (*aadA1*, *aadA2*, ant(3’’)-Ia and aph(3’’)-Ib) was the most abundant within the different antimicrobial resistance genes cassettes (ARCs) a finding that aligns with previous studies within Alberta [31,63].

Two ARCs that carried beta lactam gene with profiles, *aph(3’’)-Ib/ blaTEM-1C/ sul2* and *blaTEM-1B/ tet(R)/tet(A)* were associated with transposons which are a common category of mobile DNA element in *E. coli* and *Salmonella* spp [64]. ARC4, detected in a single clinical and four cattle isolates with the *ant(3’’)-Ia* (streptomycin) and *sul1* (sulfisoxazole) is indicative of an ARC of both cattle and human origin. ARC9, a non-integrated ARC *aph(3’’)-Ib*, *aph(6)-Id* and *sul2* in three clinical isolates may have the propensity to become integrated with an integron or transposon which can facilitate the dissemination of ARGs. Like prophages, fragmented assemblies failed to be annotated if transposons and integrons were carried on plasmids, as plasmid replicons were <1 kb. However, the majority of plasmid replicons detected in this study, were of the Inc plasmid type, especially IncF which are common plasmid replicons described in *E. coli* from animal and human sources [65]. Amongst the IncF plasmids, IncFII (clinical, *n* = 126 and cattle, *n* = 116), and IncFIB (AP001918) (clinical, *n* = 134 and cattle, *n* = 111) were equally distributed between clinical and cattle isolates, although some were unique to clinical (IncFII(pCoo), *n* = 1) and cattle (IncFIA(HI1) (*n* = 1), IncFIB(K) (*n* = 1) and IncFIC(FII) (*n* = 6)) isolates. Similar/different types of plasmid replicon in the bacterial strains of different origin may be useful in plasmid characterization and spread of ARGs, since certain plasmid types such as IncF, IncI, IncA/C and IncL are mostly associated with ARGs in *Enterobacteriaceae* [65].

## 4. Conclusions

This study analysed O157 STEC from cattle and clinical sources in the same geographic region using in silico typing methods. MLST and *stx* typing revealed that most O157:H7 isolates from cattle and clinical samples had a similar ST11 lineage and *stx1*+/*stx2*+ profile. Furthermore, a common profile for additional virulence genes (*eae*, *hylA*, *tir*, *espA*, *espB*, *espP*, *espJ*, *katP*, *toxB*, *nleABC* and *chuA*,) was observed between cattle and clinical isolates. Beta lactam resistance genes were only detected in four clinical isolates. The majority of clinical isolates with *stx2*+ profile, had a defective (nucleotide sequence deletion or insertion) *wrbA* loci which was unoccupied by *stx*2-carrying phage. There is considerable interest in understanding why there was a huge disparity in *stx*2-carrying phage occupancy in O157:H7 STEC from clinical and cattle samples in the same geographic region, which may highlight the knowledge gap between *stx*2-phage occupancy and dissemination of *stx*2. We intend to explore long read sequencing in combination with short reads to generate hybrid assemblies to further define unique features that have the potential to differentiate cattle from clinical O157:H7 isolates. If this mosaic in *stx*2-carrying phage occupancy between the clinical and cattle isolates is a stable trait, it could be employed for food safety screening and monitoring of STEC O157:H7 in both farm and clinical environments.

## 5. Materials and Methods

### 5.1. Sample Collection and Bacterial Isolation

Cattle faecal samples were collected from 2007 to 2015 from commercial feedlot pens or from the floors of transport trucks. At feedlots, *E. coli* were isolated from rectal grab samples, hide swabs and pooled fecal pats from the pen floor (Appendix A). Cattle isolates were collected from nine locations in southern Alberta, Canada.

For cattle isolates, presumptive O157 *E. coli* [66,67] were retrieved from storage at −80 °C, thawed and 1 mL incubated in 4 mL EC broth at 37 °C for 24 h. A 1 mL aliquot of the enriched culture was then centrifuged at 8000× *g* for 10 min before extraction of DNA from the pellet using the NucleoSpin Tissue Kit (Macherey-Nagel, Islington, ON, Canada). Extracted genomic DNA was used to confirm O157 serogroup and the presence of *stx*1, *stx*2 and *eae* using primers and PCR as described by Conrad et al. [68].

Human clinical O157:H7 isolates (322) were isolated from 2007 to 2015 from stools cultures from patients experiencing gastroenteritis from Alberta and O157 was confirmed by antiserum agglutination [69,70,71]. A random subset (*n* = 127) were selected and sequenced for the same sampling years as the cattle isolates. DNA was extracted using the MagaZorb DNA mini-prep kit (Promega Corporation, Madison, WI, USA). The University of Calgary Conjoint Health Research Ethics Board approved the study, REB19–0510.

### 5.2. Bacterial Whole Genome Sequencing and Data Analysis

DNA was extracted from 252 bacterial isolates collected from cattle (*n* = 125) and humans (*n* = 127) and were subjected to whole genome sequencing on an Illumina NovaSeq 6000 at Génome Québec (Montreal, QC, Canada) to generate 250 bp paired-end reads. Sequence data were downloaded as Fastq files and assessed for quality using the FASTQC tool. Trimmomatic v0.36.5 was used to remove adapter sequences as well as low quality sequences based on a Phred Q score of <30 (99.9%). Reads that met quality standards were de novo assembled into contigs using the Unicycler pipeline [72] and annotated using Prokka [73]. Using an average genome size of 5.5 Mb for *E. coli* O157, the average sequencing coverage was estimated at 158x. Assemblies of all 252 isolates generated, on average, 276 contigs per genome with an average of 117 contigs being ≥1kb size, and N50 contig length of 154,553 bases (Appendix A). The average genome size of the sequenced isolates was 5,304,568 bp (5.3 Mb). The assembled contigs were used for in silico analysis as described. The draft whole genome sequence assemblies of the 252 bacterial isolates have been deposited in GenBank under BioProject ID PRJNA870153.

### 5.3. In Silico Serotyping and Multi-Locus Sequence Typing (MLST) Analysis

The ECTyper tool for *Escherichia coli* serotyping, version 1.0. [74] was used to confirm the serotype of isolates as O157 using default parameters: O antigen minimum ≥ 90% identity/coverage and H antigen minimum ≥ 90% identity and ≥50% coverage. Genetic relatedness of cattle and clinical O157 STEC strains were determined using an in silico *E. coli* MLST scheme. Seven housekeeping genes loci (*adk*, *fumC*, *gyrB*, *icd*, *mdh*, *purA*, and *recA*), previously described for *E. coli* [27] were used in MLST. The *E. coli* MLST database was used to assign a number to each locus and a sequence type (ST) for each unique combination of loci.

### 5.4. Identification of *Stx* Genes, Prophages, and *Stx*-Carrying Phage and Associated-Insertion Sites

Chromosomal sequences of O157 isolates were verified for *stx* genes and subtypes by searching genomes using the *stx*1 and *stx*2 primers [68], followed by subtype (*stx*1a-*stx*1d and *stx*2a-*stx*2f) primers [75] using Geneious 10.2.6. FASTA files of O157 genomes were queried for intact, questionable, and incomplete prophages harbored within the strains using Phage Search Tool Enhanced Release [76]. Hits with a query > 70% were considered as incomplete, 70 to 90% as questionable and >90 as intact prophages [76]. The *stx*-carrying contigs and flanking genes were manually screened for prophage carrying *stx* with Geneious 10.2.6 using *E. coli* O157:H7 Sakai phage (Sp5 and Sp15) as a reference. The sequences for *wrbA*, *mlrA*, *argW*, *sbcB*, and *yecE* from *E. coli* O157:H7 strain Sakai were manually curated against assembled contigs of O157 isolates to identify *stx* insertion sites with possible occupancy based on the presence of the *stx*-phage integrase (*int*) using Geneious. To understand the variability between the presence of *stx*-associated phage and unoccupied *wrbA* or *mlrA* insertion sites, we aligned the sequence of the *wrbA* and *mlrA* region in cattle and clinical isolates against *wrbA* and *mlrA* DNA sequence from O157 strain Sakai using a pairwise sequence alignment tool, in the EMBL-EBI search and sequence analysis tools [77].

### 5.5. In Silico Determination of Virulence Factors, Antimicrobial Resistance, and Mobile Element Genes

VirulenceFinder 2.0 was used for the identification of virulence genes in each of the assembled contigs with default settings: coverage ≥ 60% and identity ≥ 90% (https://cge.cbs.dtu.dk/services/VirulenceFinder, accessed on 2 February 2022) [78]. Assembled genomes were screened against ResFinder antimicrobial resistance database (https://bitbucket.org/genomicepidemiology/resfinder_db.git, accessed on 15 December 2021) to determine the presence of antimicrobial resistance genes (ARGs) and PlasmidFinder; (https://bitbucket.org/genomicepidemiology/plasmidfinder_db.git, accessed on 15 December 2021) to detect plasmids. ARGs and/or antimicrobial resistance genes cassettes (ARCs; identical resistance/flanking gene profile in different STEC isolates) flanked by mobile element gene (Integron/transposon) were manually identified using Geneious.

## Figures and Tables

**Figure 1 toxins-14-00603-f001:**
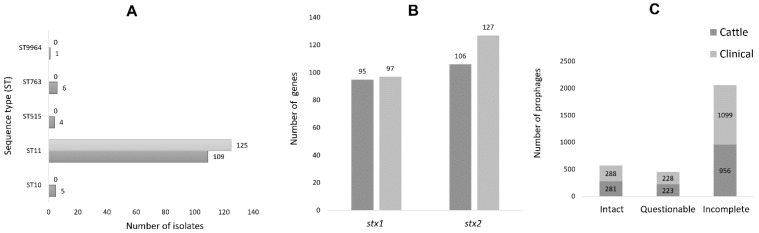
(**A**) Genetic relatedness of O157 STEC using MLST scheme with seven housekeeping genes: adk, fumC, gyrB, icd, mdh, purA, and recA, (**B**) the number of identified *stx* genes and (**C**) prophages in cattle and clinical O157 STEC isolates.

**Figure 2 toxins-14-00603-f002:**
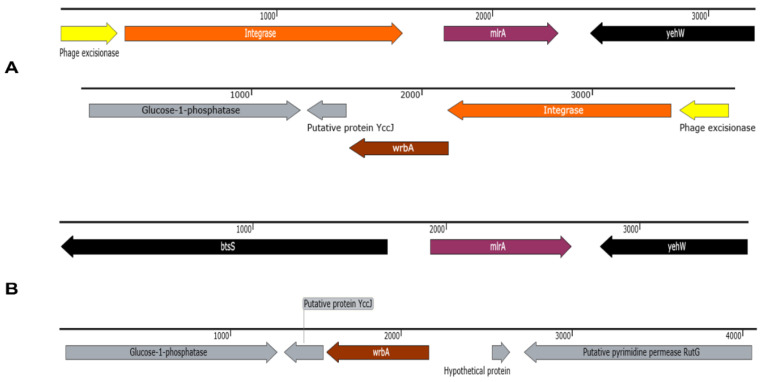
(**A**) Presence of *stx*-carrying prophage integrase flanking *mlrA* and *wrbA* in cattle and clinical O157 STEC and (**B**) absence of *stx*-carrying prophage integrase at *mlrA* and *wrbA* in cattle and clinical O157 STEC. Arrow color indicates gene direction.

**Figure 3 toxins-14-00603-f003:**
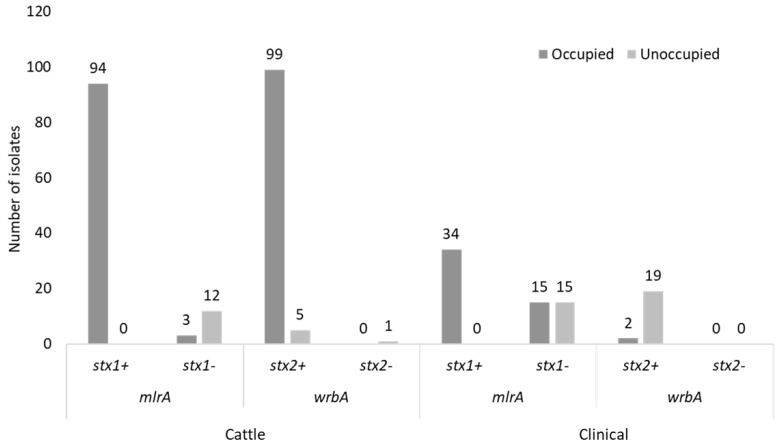
Distribution of *stx* genes in cattle and clinical O157 STEC isolates with occupied and unoccupied insertion (*mlrA* and *wrbA*) regions.

**Figure 4 toxins-14-00603-f004:**
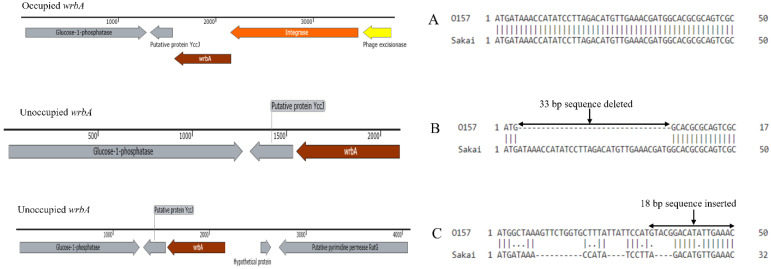
Sequence alignment for occupied and unoccupied *wrbA* gene of O157 isolates with that of *E. coli* O157 strain Sakai. Only the first 50 bp are represented (**A**) no polymorphism, (**B**) 33 bp deleted in O157 and (**C**) 18 bp inserted in O157, compared to Sakai strain.

**Figure 5 toxins-14-00603-f005:**
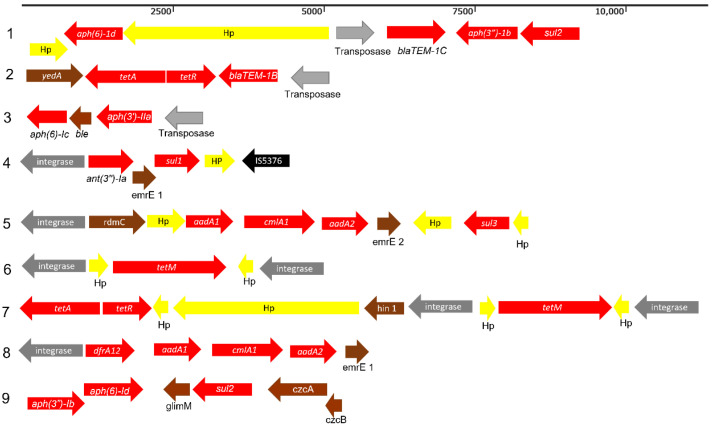
Identified antibiotic resistance cassettes (ARCs). Arrow color indicates gene direction. Gray; mobile gene element, Red; resistance gene, Yellow; hypothetical protein, Black; insertion sequence, Brown; other gene.

**Table 1 toxins-14-00603-t001:** Predicted ARGs profiles and virulence genes for cattle and clinical isolates.

Number of Isolates	Source	Year	**stx*1*	**stx*2*	Resistance Genotype
4	Cattle	2014	−	+	*ant(3’’)-Ia, sul1, tet(A)*
1	Clinical	2009	+	+	
1	Clinical	2008	+	+	*blaTEM-1B*
3	Clinical	2014	−	+	*aph(6)-Id, aph(3’’)-Ib, sul2, tet(B)*
1	Clinical	2015	+	+	*blaTEM-1B, tet(A)*
1	Clinical	2007	−	+	*aadA1, aadA2, cmlA1, sul3*
1	Clinical	2014	−	+	*aadA2, ant(3’’)-Ia, cmlA1, sul3, tet(A)*
1	Clinical	2009	+	+	*aph(3’’)-Ib, aph(6)-Id, blaTEM-1C, sul2*
1	Clinical	2009	+	+	*aph(3’’)-Ib, aph(6)-Id, blaTEM-1B, sul2, tet(B)*
1	Clinical	2015	−	+	*aph(3’’)-Ib, aph(6)-Id, dfrA14, floR, sul2, tet(A)*
1	Cattle	2014	−	−	*aadA1, aadA2, aph(3’)-IIa, aph(6)-Ic, cmlA1, dfrA12, sul3, tet(A), tet(B), tet(M)*
1	Cattle	2013	+	+	*aph(3’’)-Ib, aph(6)-Id, floR, sul2, tet(A)*
1	Cattle	2009	−	−	*tet(M)*
3	Cattle	2015	−	−	*tet(A), tet(M)*
1	Cattle	2015	−	−	*tet(A)*

Streptomycin; (*aadA1, aadA2,*
*ant(3’’)-Ia, aph(6)-Ic, aph(3’’)-Ib), Kanamycin; (aph(3′)-IIa, aph(6)-Id*), Trimethoprim; (*dfrA14*), Chloramphenicol; (*cmlA1, floR*), Sulfisoxazole; (*sul2*), Tetracycline; (*tetA, tetE, tetM,*), Ampicillin; (*blaTEM-1B*). +; gene present, −; gene absent.

## Data Availability

Data of the 252 bacterial isolates generated in this study have been deposited in GenBank under BioProject ID PRJNA870153.

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
