# Peer review of "Genomic Analysis of Shiga Toxin-Producing E. coli O157 Cattle and Clinical Isolates from Alberta, Canada"

_toxins, 2022, doi:10.3390/toxins14090603_

Round 1

Reviewer 1 Report

I think the manuscript entitled “Genomic analysis of Shiga toxin-producing E. coli O157 cattle and clinical isolates from Alberta, Canada” is very interesting. Manuscript is well written and designed. I think the manuscript justified its title and gives a brief idea about the study. However, authors need to focus on typo errors and correct them before the final submission.

For example, “E. coli” must be in italics, unlike in L95, L96, L129 and so on…

1.       Authors should also brief about the harmful disease spreading upon infection with E. coli O157.

2.       I would also encourage you to write a few lines about how Shiga toxins induce toxicity and why it is important to identify the type of bacteria which are producing this toxin.

3.       What is the half-life of Shiga toxins in blood?

4.       Please suggest some therapeutic strategies to prevent the toxicity incused due to the E. coli O157 infection.

5.       What are the other bacteria isolated from Cattle faecal samples?

6.       It is unclear for the isolation of E. coli O157 from humans which samples were used?

7.       How have authors confirmed and identified the bacteria? What is the percentage of sequence similarities? Was the raw sequence data submitted in any data bank? Please provide the accession ID of the submitted sequence for public access.

8.       I strongly recommend modifying the introduction section of the manuscript. The below mentioned papers are suitable for citation:

Joseph et al., Toxins (Basel). 2020 Jan 21;12(2):67.

Dey et al., Microbiol Res. 2020 Aug;237:126489.

Cheng et al., Toxins (Basel). 2013 Oct 22;5(10):1845-58.

Author Response

Reviewer One

Comments and Suggestions for Authors

I think the manuscript entitled “Genomic analysis of Shiga toxin-producing E. coli O157 cattle and clinical isolates from Alberta, Canada” is very interesting. Manuscript is well written and designed. I think the manuscript justified its title and gives a brief idea about the study. However, authors need to focus on typo errors and correct them before the final submission.

For example, “E. coli” must be in italics, unlike in L95, L96, L129 and so on

Response: All gene names have been italicized

  1. Authors should also brief about the harmful disease spreading upon infection with E. coli O157.

Response: Disease occurrence upon infection with E. coli O157 such as bloody diarrhoea, thrombocytopenia, haemorrhagic colitis and hemolytic uremic syndrome has been included. (Lines 64 – 68)

  1. I would also encourage you to write a few lines about how Shiga toxins induce toxicity and why it is important to identify the type of bacteria which are producing this toxin.

Response: Thanks, Shiga toxin-producing strains other than E. coli, such as Shigella spp have been added in the introduction section. The importance of identifying these species has been included in the introduction as well (lines 59 – 63)

3.What is the half-life of Shiga toxins in blood?

Response: The Shiga toxin half life is about 5 days in human blood Brigotti et al, 2006. This information is included in the text (Line 69)

  1. Please suggest some therapeutic strategies to prevent the toxicity incused due to the E. coli O157 infection.

Response: Possible therapeutic strategies such as DNA-based vaccines, probiotics and polyphenolic compounds have been suggested (Lines 74 – 80)

  1. What are the other bacteria isolated from Cattle faecal samples?

Response: Shigella flexneri and S. sonnei, which are also responsible for diarrhoea and hemolytic uremic syndrome, and can be isolated from cattle feces have been included in the introduction section (lines 59 – 60).

  1. It is unclear for the isolation of E. coli O157 from humans which samples were used?

Response: E. coli O157 from clinical samples was cultured from stools from patients experiencing gastroenteritis. This information has been added in the material and methods section (lines 348 – 349).

  1. How have authors confirmed and identified the bacteria?

Response: The serogroup primers specific for O157 (Conrad, et al 2014) and antiserum agglutination test (Couturier et al, 2011) were used to confirm cattle and clinical isolates, respectively. Bacterial identity was further confirmed using the ECTyper tool for E coli serotyping. It is already stated in the material and methods section (line 365).

What is the percentage of sequence similarities? Was the raw sequence data submitted in any data bank? Please provide the accession ID of the submitted sequence for public access.

Response: Comparative phylogenomic analysis using the whole genome sequence data is in progress and will be published as a separate article. However, comparative analysis of stx1 and stx2 genes revealed their 100% sequence identity across all isolates carrying these genes. This information is now included in the results (line 122).  The sequence data for these isolates is being uploaded to the NCBI whole genome sequence database (GenBank) and will be made available for public access under BioProject ID  PRJNA870153 upon publication of this manuscript. This information is included in the manuscript text (lines 364-365).

  1. I strongly recommend modifying the introduction section of the manuscript. The below mentioned papers are suitable for citation:

Response: Thanks, the introduction section has been rewritten and includes the following: disease caused by STEC infection, possible therapeutic and preventive measures.

Joseph et al., Toxins (Basel). 2020 Jan 21;12(2):67.

Dey et al., Microbiol Res. 2020 Aug;237:126489.

Cheng et al., Toxins (Basel). 2013 Oct 22;5(10):1845-58.

Response: References have been added in the modified introduction section.

Reviewer 2 Report

This study compared STEC O157 isolates from cattle and clinical sources in the same geographic region using whole genome sequencing, and revealed that most O157:H7 isolates from cattle and clinical samples had a similar ST11 lineage and virulence genes profile. The main finding is the huge disparity in stx2-carrying phage occupancy in O157:H7 STEC from clinical and cattle samples in the same geographic region.

 Main concern: Some isolates from cattle were O157:non-H7 and absence of stx1/stx2 genes. These isolates should not be defined as “Shiga toxin-producing E. coli”. It would be bias when these isolates were included in comparison to clinical isolates.

 Gene name should be italic throughout the manuscript.

Author Response

Reviewer Two

Comments and Suggestions for Authors

This study compared STEC O157 isolates from cattle and clinical sources in the same geographic region using whole genome sequencing, and revealed that most O157:H7 isolates from cattle and clinical samples had a similar ST11 lineage and virulence genes profile. The main finding is the huge disparity in stx2-carrying phage occupancy in O157:H7 STEC from clinical and cattle samples in the same geographic region.

 Main concern: Some isolates from cattle were O157:non-H7 and absence of stx1/stx2 genes. These isolates should not be defined as “Shiga toxin-producing E. coli”. It would be bias when these isolates were included in comparison to clinical isolates.

Response: Thank, we have indicated in the results section that the O157:non-H7 (n = 16) lacked stx1/stx2 genes and also stated in the discussion section that they were considered as non-STEC to differentiate these strains from STEC strains.

 Gene name should be italic throughout the manuscript.

Response: All gene names have been italicized

Round 2

Reviewer 1 Report

The current revised manuscript has been widely revised keeping the reviewer's comment in mine. The reply to the comments are satisfactory to me. I believe the present format of the manuscript can be accepted. 

Author Response

Thanks for you recommendation.  

Reviewer 2 Report

1. Though the authors have indicated in the results section that the O157:non-H7 (n = 16) lacked stx1/stx2 genes and also stated in the discussion section that they were considered as non-STEC to differentiate these strains from STEC strains, it should also be noted where percentage used to compare the difference between cattle and clinical isolates.

2. Gene name should be italic throughout the manuscript. Some are still there, such as L146, L151, etc.

Author Response

Comments and Suggestions for Authors

  1. Though the authors have indicated in the results section that the O157:non-H7 (n = 16) lacked stx1/stx2 genes and also stated in the discussion section that they were considered as non-STEC to differentiate these strains from STEC strains, it should also be noted where percentage used to compare the difference between cattle and clinical isolates.

Response: Non-STEC strains have been stated to differentiate from STEC strains in the comparative analysis (lines 136-137 and 177-178).

  1. Gene name should be italic throughout the manuscript. Some are still there, such as L146, L151, etc.

Response: Gene names have been italicized, we believe we had done this in the previous version as well and are wondering if some of these are being lost during manuscript upload and conversion.